# An Integrated In Silico, In Vitro and Tumor Tissues Study Identified Selenoprotein S (SELENOS) and Valosin-Containing Protein (VCP/p97) as Novel Potential Associated Prognostic Biomarkers in Triple Negative Breast Cancer

**DOI:** 10.3390/cancers14030646

**Published:** 2022-01-27

**Authors:** Susan Costantini, Andrea Polo, Francesca Capone, Marina Accardo, Angela Sorice, Rita Lombardi, Palmina Bagnara, Federica Zito Marino, Martina Amato, Michele Orditura, Maddalena Fratelli, Gennaro Ciliberto, Alfredo Budillon

**Affiliations:** 1Experimental Pharmacology Unit, Laboratori di Mercogliano, Istituto Nazionale Tumori-IRCCS-Fondazione G. Pascale, 80131 Napoli, Italy; a.polo@istitutotumori.na.it (A.P.); f.capone@istitutotumori.na.it (F.C.); angy.sorice@virgilio.it (A.S.); r.lombardi@istitutotumori.na.it (R.L.); palminabagnara93@gmail.com (P.B.); 2Pathology Unit, Department of Mental and Physical Health and Preventive Medicine, Università degli Studi della Campania “Luigi Vanvitelli”, 80131 Napoli, Italy; marina.accardo@unicampania.it (M.A.); federica.zitomarino@unicampania.it (F.Z.M.); martina.amato@unicampania.it (M.A.); 3Division of Medical Oncology, Department of Precision Medicine, School of Medicine, Università degli Studi della Campania “Luigi Vanvitelli”, 80131 Napoli, Italy; michele.orditura@unicampania.it; 4Pharmacogenomics Unit, Istituto di Ricerche Farmacologiche Mario Negri, IRCCS, 20156 Milan, Italy; maddalena.fratelli@marionegri.it; 5IRCCS National Cancer Istituto Regina Elena, 00144 Roma, Italy; gennaro.ciliberto@ifo.gov.it

**Keywords:** selenoproteins, biomarkers, SELENOS, VCP/p97, breast cancer, TNBC, bioinformatics analysis

## Abstract

**Simple Summary:**

Triple negative breast cancer (TNBC) represents a clinical challenge because its early relapse, poor overall survival and lack of effective treatments. Altered levels selenoproteins have been correlated with development and progression of some cancers, however, no consistent data are available about their involvement in TNBC. Here we analyzed the expression profile of all twenty-five human selenoproteins in TNBC cells and tissues by a systematic approach, integrating in silico and wet lab approaches. We showed that the expression profiles of five selenoproteins are specifically dysregulated in TNBC. Most importantly, by a bioinformatics analysis, we selected selenoprotein S and its interacting protein valosin-containing protein (VCP/p97) as inter-related with the others and whose coordinated over-expression is associated with poor prognosis in TNBC. Overall, we highlighted two mechanistically related novel proteins whose correlated expression could be exploited for a better definition of prognosis as well as suggested as novel therapeutic target in TNBC.

**Abstract:**

Background. Triple negative breast cancer (TNBC) is a heterogeneous group of tumors with early relapse, poor overall survival, and lack of effective treatments. Hence, new prognostic biomarkers and therapeutic targets are needed. Methods. The expression profile of all twenty-five human selenoproteins was analyzed in TNBC by a systematic approach.In silicoanalysis was performed on publicly available mRNA expression datasets (Cancer Cell Line Encyclopedia, CCLE and Library of Integrated Network-based Cellular Signatures, LINCS). Reverse transcription quantitative PCR analysis evaluated selenoprotein mRNA expression in TNBC versus non-TNBC and normal breast cells, and in TNBC tissues versus normal counterparts. Immunohistochemistry was employed to study selenoproteins in TNBC tissues. STRING and Cytoscape tools were used for functional and network analysis. Results.GPX1, GPX4, SELENOS, TXNRD1 and TXNRD3 were specifically overexpressed in TNBC cells, tissues and CCLE/LINCS datasets. Network analysis demonstrated that SELENOS-binding valosin-containing protein (VCP/p97) played a critical hub role in the TNBCselenoproteins sub-network, being directly associated with SELENOS expression. The combined overexpression of SELENOS and VCP/p97 correlated with advanced stages and poor prognosis in TNBC tissues and the TCGA dataset. Conclusion. Combined evaluation of SELENOS and VCP/p97 might represent a novel potential prognostic signature and a therapeutic target to be exploited in TNBC.

## 1. Introduction

Triple-negative breast cancer (TNBC) is a heterogeneous group of tumors, more prevalentin younger women and characterized by early relapse, poor overall survival and the lack of effective treatments, currently based mainly on chemotherapy, thus representing a clinical challenge [1]. Gene expression and mutation profiling have suggested target therapies in defined subgroups of patients and some positive results were observed with immunotherapy approaches [2,3]. However, a large number of TNBC patients fail to demonstrate a good response to such treatments. Novel prognostic and predictive biomarkers to improve clinical progress are needed.

Selenium is an essential micronutrient for humans and other mammals, and it is involved in different molecular processes that have an antioxidant, anti-inflammatory and anti-viral role. Selenium biological effects are exerted through several selenium-containing proteins, known as selenoproteins, containing selenocysteine (Sec), an analogue of cysteine with a selenol group replacing the sulfur-containing thiol group [4]. Until now, twenty-five selenoproteins have been identified in humans, subdivided into two groups accordingto the position of selenocysteine in the sequence. The first group includes five glutathione peroxidases (GPX1, 2, 3, 4 and 6), three iodothyronine deiodinases (DIOs) and eight other selenoproteins (SELENOF, SELENOH, SELENOM, SELENON, SELENOT, SELENOV, SELENOW and SEPHS2). The second group belongs to three thioredoxin reductases (TXNRDs) and six other selenoproteins (methionine-R-sulfoxidereductase 1 (MSRB1), SELENOI, SELENOK, SELENOO, SELENOP and SELENOS) [5].

The role of selenoproteins in carcinogenesis, as well as their mechanism of action and regulation, remains ambiguous and certainly needs further clarification. Indeed, although many selenoproteins are dysregulated in cancer cells and tissues, depending on the cancer types and setting, they could be either overexpressed or downregulated, and their functions, prevalently in the regulation of oxidative homeostasis, could be either detrimental or favor cancer initiation and/or progression. Thus, it is not clear how and if these proteins can be used as clinical diagnostic/prognostic markers. In detail, selenoproteins, by their antioxidant activity, are able to block cancer development by inducing a slowdown of the oxidative insult and the related DNA damage [6]. However, selenoproteins can be implicated in both oncogenic and tumor-suppressive pathways, such as PI3K/Akt/mTOR, c-Met, MAPK and VEGF [7,8,9], thus being involved in all hallmarks of cancer [6]. Indeed, selenoproteins play crucial roles in metastatic processes, including matrix degradation and migration, cell adhesion and invasion in the blood and extravasations into secondary tissues [10]. Furthermore, it is also important to underline that some selenoproteins, including SELENOF, SELENOK, SELENOM, SELENON, SELENOS, SELENOT and DIO2, are localized in the endoplasmic reticulum (ER) organelle and involved in protein degradation and regulation of ER stress that has been observed in many types of cancer cells, being associated with cell survival and resistance to anticancer drugs [11].

Recently, to better define the correlation between the selenoprotein family and cancer, we took advantage of a bioinformatics approach and studied the interaction network between all twenty-five selenoproteins, highlighting the presence of six HUB nodes (ABL1, EP300, FYN, MYC, PSMB2 and SRPK2) that play the strongest role in coordinating the obtained network. Within this analysis, we also analyzed the correlation between some selenoproteins and/or HUB node gene expression and patient outcome in ten solid tumors [12]. Moreover, we evaluated the gene expression levels of the twenty-five selenoproteins and of the six HUB nodes identified in four prostate cancer cell lines compared to normal prostate epithelial cells, selecting three selenoproteins (GPX2, EP300 and PSMB2), whose collective expression correlated with the overall survival of prostate cancer patients, suggesting that only the combined evaluation of some selenoproteins and HUB nodes could have prognostic value and may improve patient outcome prediction [12].

In breast cancer, several studies demonstrated that low serum selenium levels correlated with cancer initiation [13]. Moreover, an overexpression of DIO1 was found in breast cancer tissues and associated with advanced stages [14]. Similarly, an overexpression of TXNRD1 in human breast cancer was indicated as an index of cancer progression and associated with poor prognosis [15,16]. Some reports showed that genetic variations in some selenoproteins, such as GPX1, GPX3, GPX4, SELENOS, and TXNRD1, are correlated to breast cancer development [17,18,19].

Our group performed a preliminary analysis in two human breast cancer cell lines demonstrating that the mRNA expression of some selenoproteins was dysregulated among the twenty-one examined [20]. More recently, we evaluated SEPHS2 expression in TNBC tissues compared to normal counterparts, showing the overexpression of this selenoprotein in TNBC associated with higher tumor grading [21].

However, no detailed studies have been reported so far about the expression levels of all twenty-five selenoproteins in the different breast cancer subtypes.

Valosin-containing protein (VCP/p97) is an ATPase that governs many cellular processes, ranging from the degradation of unfolded proteins to chromatin regulation and other key cellular events, representing an attractive anticancer drug target [22]. Recently, high expression levels of VCP/p97 were found to be involved in colorectal cancer growth, invasion and metastasis through STAT3 signaling activation [23]. VCP/p97was also correlated with stage, grading, recurrence rate and shorter overall survival in B-cell lymphoma patients [24].

Notably, during the ER-associated protein degradation (ERAD) process, the recruitment of VCP/p97 to the ER membrane is essential for substrate degradation and it is mediated by selenoprotein SELENOS [25,26].

Here, by anin silicoandin vitrogene expression analysis, we evaluated all twenty-five selenoproteins in TNBC cell lines compared to other breast cancer subtypes. We next evaluated the gene expression for all selenoproteins in TNBC tissues compared to adjacent normal breast tissue counterparts. Functional analyses and interaction network studies aimed to identify HUB node(s) more correlated to selenoproteins, evidenced that VCP/p97 plays an important role in the interaction sub-network of selenoproteins in TNBC, being directly related with SELENOS. Consequently, by evaluating SELENOS and VCP/p97 in TNBC cell lines, tumor tissues, and the TCGA dataset, we evidenced their correlated expression, suggesting that both proteins might represent novel potential prognostic biomarkers and/or therapeutic targets in TNBC.

## 2. Materials and Methods

The expression profile of all twenty-five human selenoproteins was analyzed in TNBC by a systematic approach including:in silicoanalysis of publicly available datasets; reverse transcription real-time quantitative PCR (RT-qPCR) and western blotting on cell lines; and RT-qPCR and immunohistochemistry (IHC) on tumor tissues. Functional analysis and network interaction studies were performed by different bioinformatics for in-depth interpretation of the data.

### 2.1. In Silico Analysis of Gene Expression Profiles of Selenoproteins

The mRNA expression data of twenty-five selenoproteins, obtained by the RNAseq method, in fifty-seven breast cancer cell lines were retrieved from the Cancer Cell Line Encyclopedia (CCLE), a comprehensive collection of genomic data by massively parallel sequencing from 1457 human cancer cell lines [27]. Moreover, we extrapolated the mRNA expression data of seventeen selenoproteins on 30 breast cell lines from the Library of Integrated Network-based Cellular Signatures (LINCS) dataset, called “Breast Cancer Profiling Project, Gene Expression 1: Baseline mRNA sequencing” (http://lincs.hms.harvard.edu/db/datasets/20348/main, accessed on 3 March 2020). Breast cancer cell lines were classified in “TNBC” and “non-TNBC” according to data reported in [28,29], ATCC collection (https://www.lgcstandards-atcc.org/en.aspx, accessed on 3 March 2020), Cellosaurus resource [30], DSMZ-German Collection of Microorganisms and Cell Cultures–DSMZ (https://www.dsmz.de/dsmz, accessed on 3 March 2020), and literature [31]. Clustering analysis was performed by MetaboAnalyst (https://www.metaboanalyst.ca/, accessed on 3 December 2021) and by script in house using the Complex Heatmap in R package.

The UALCAN database was used to obtain plots depicting the expression profile of selenoproteins and hub nodes in TNBC tissues by the breast invasive carcinoma TCGA dataset [32].

Using the PROGgeneV2 online tool [33], we evaluated the correlation between SELENOS and VCP/p97 gene expression and overall survival in TNBC samples collected in TCGA_BRCA (Breast Cancer).

### 2.2. Cell Lines

Normal epithelial breast cell line (MCF-10A), two TNBC cell lines (MDAMB231 and MDAMB468), three estrogen receptor (ER) and progesterone receptor (PR) positive cell lines (HCC1500, MCF7 and MDAMB175), two HER2 positive cell lines (SKBR3 and SUM185), and two ER/HER2 and PR/ER/HER2 positive cell lines (HCC1419 and MDAMB361) are all from the American Type Culture Collection (ATCC, Rockville, MD, USA).

MCF-10A cells were grown in DMEM supplemented with 20ng/mL human epidermal growth factor (Life Technologies, Carlsbad, CA, USA), 10 μg/mL human insulin (Life Technologies, Carlsbad, CA, USA), and 0.5 μg/mL of hydrocortisone (Sigma-Aldrich, St. Louis, MO, USA).

MDA-MB231 cells were grown in RPMI 1640 (Lonza, Muenchensteinerstrasse, Basel, Switzerland) supplemented with penicillin/streptomycin (100×) (Euroclone, Pero, Milan, Italy), fetal bovine serum (10%) (Invitrogen, Waltham, MA, USA), and Glutamax (100×) (Invitrogen, Waltham, MA, USA). MDA-MB468 cells were grown in DMEM/Ham’s F-12 50/50 supplemented with fetal bovine serum (10%) (Invitrogen, Waltham, MA, USA), penicillin/streptomycin (100×) (Euroclone, Pero, Milan, Italy) and Glutamax (2 mM) (Invitrogen, Waltham, MA, USA).

MCF7, SKBR3 and SUM185 cells were grown in DMEM supplemented with fetal bovine serum (10%) (Invitrogen, Waltham, MA, USA), penicillin/streptomycin (100×) (Euroclone, Pero, Milan, Italy) and Glutamax (100×) (Invitrogen, Waltham, MA, USA).

MDAMB175, MDAMB361, HCC1419 and HCC1500 cells were grown in DMEM/Ham’s F-12 50/50 supplemented with fetal bovine serum (5%) (Invitrogen, Waltham, MA, USA), penicillin/streptomycin (100×) (Euroclone, Pero, Milan, Italy) and Glutamax (2mM) (Invitrogen, Waltham, MA, USA).

Cultures were maintained in a humidified atmosphere of 95% air and 5% CO_2_ at 37 °C. All cell lines were regularly inspected for mycoplasma. The cells were authenticated with a short tandem repeat profile generated by LGC Standards (Sesto San Giovanni, Milan, Italy).

### 2.3. RNA Preparation and RT-qPCR

To extract total RNA from ten cell lines, we used an RNAeasy Mini Kit (Qiagen Inc., Germantown, MD, USA). For the tissue samples (see below), we removed the paraffin using xylene extraction for RNA isolation, and, to extract total RNA from tissue sections equivalent to 60 µm (three 20 µm sections), we used the Recoverall (TM) Total RNA Isolation Kit (Life Technologies, Carlsbad, CA, USA) according to the manufacturer’s instructions. The RNA concentration and purity were determined using the NanoDrop 2000 spectrophotometer (Thermo Scientific, Wilmington, DE, USA) at 260/280 nm of optical density. Reverse transcription of RNA was performed with 2 μg of RNA using a SuperScript VILO cDNA Synthesis kit (Life Technologies, Carlsbad, CA, USA) in a 20 μL reaction volume.

Sequence for mRNA from the nucleotide data bank (NCBI, Bethesda, MD, USA) was used to design primer pairs for RT-qPCR with an amplicon <100 bp according to the manufacturer’s instructions. Oligonucleotides were obtained from Eurofins Technologies (Budapest, Fótiút, Ungheria). The list of primers is reported in Appendix A. RT-qPCR experiments were performed using the Step-One Real Time PCR System (Applied Biosystems, Waltham, MA, USA). Each aliquot of cDNA (2 µL) was amplified in a mixture (25 µL) consisting of the reverse and forward primers (300 nM) and 2X SYBR Green PCR Master Mix (Applied Biosystems, Waltham, MA, USA). The conditions used for PCR were 5 min of denaturation at 95 °C, followed by 44 cycles of a two-step program: (i) denaturation at 95 °C for 30 s; and (ii) annealing/extension at 60 °C for 1 min. Each assay included a no-template control for each primer pair. Each reaction was repeated at least three times. β-Actin mRNA was used to normalize the data. All obtained data were analyzed statistically. Sample ΔCq values were calculated as the difference between the mean Cq obtained for each selenoprotein transcript (seleno-mRNA) and housekeeping gene in the experiments on cells or tissues. The 2^−ΔΔCq^ values were determined in order to define the fold change of expression level for each seleno-mRNA in different breast cancer cells compared to the non-cancerous MCF-10A cells, and in TNBC tissues compared to normal breast counterparts. Raw data related to gene expression analyses are available at https://gbox.garr.it/garrbox/index.php/s/he1ukq7pDNJ45WP uploaded on 1 July 2021.

### 2.4. Western Blot Analysis

Western blot analysis experiments were performed as previously described [34]. In brief, total protein extracts from cell lines were dissolved in a lysis buffer and the protein concentration was measured by the Bradford assay. Fifty micrograms of protein lysates were loaded per lane, separated using 10% PAGE and transferred to PVDF membranes (Millipore Merck, Milano, Italy). After incubation with a specific primary antibody and probed with the appropriate horseradish peroxidase–linked IgG secondary antibody, immunoreactive bands were detected by ECL with an Image-Quant Las 500 instrument (GE Healthcare, Chicago, IL, USA).Primary antibodies were purchased as follows: VCP/p97 (#2649) from Cell Signaling Technology (Leiden, The Netherlands); SELENOS (#16333) from Sigma-Aldrich (St. Louis, MO, USA); β-actin (sc-47778) from Santa Cruz Biotechnology Inc. (Dallas, TX, USA). The antibodies were diluted 1:1000 in 5% nonfat dry milk (Euroclone, Pero, Milan, Italy), 1× TBS (Euroclone, Pero, Milan, Italy) and 0.1%Tween (Euroclone, Pero, Milan, Italy) at 4 °C. Secondary antibody was purchased as follows: polyclonal Goat Anti-Rabbit IgG Antibody (H+L) HRP-conjugated (#bs-0295G) from Bioss Antibodies Inc. (Woburn, MA, USA). Densitometric analysis was performed by ImageJ software [35] and protein expression levels were normalized to Ponceau Red staining, as previously reported [36].

### 2.5. Tissue Samples

A formalin-fixed paraffin-embedded (FFPE) block relative to tumor tissues obtained by surgical resection from the 30 TNBC patients was subjected to reverse transcription (RT)-qPCR, as described above, and IHC. The patients were recruited by the “University of Campania”, Naples, Italy, and their tumor tissues were the object of a recent paper from our group [21]. All patients provided written informed consent for the use of tissue samples according to the institutional regulations and the approval by the Ethics Committee of the University of Campania. Histopathological diagnoses were reviewed on standard H&E-stained slides by two co-author pathologists (MA and FZM). The clinic-pathological characteristics of the patients are described below and summarized in Table 1.

For IHC analysis tissue sections were inserted, after alcohol rehydration and xylene dewaxing, in jars containing trisodium citrate solution (0.01 M). Then, they were microwaved, rinsed for 5 min in cool H_2_O and for 30 min at room temperature in H_2_O_2_ (3%). Successively, the sections were washed in TRIS-buffered saline and incubated at 4 °C overnight with rabbit anti-SELENOS antibody (SAB2102105, Sigma-Aldrich, St. Louis, MO, USA), diluted 1:100, and rabbit anti-VCP antibody (GTX101089, GeneTex, Irvine, CA, USA), diluted 1:150. After incubation, biotinylated secondary antibodies plus streptavidin (Dako, Glostrup, Denmark) were used to stain the samples using DAB chromogen (Dako, Glostrup, Denmark) as a substrate. Mayer’s Hematoxylin solution was used as a nuclear counterstaining.

Immunoreactivity was evaluated blinded with independent assessment by two pathologists (MA and FZM) as positive stained cancer cellular percentage and staining intensity [37]. The pathologists then agreed on the final scores, indicating a weak staining with 1, a moderate staining with 2, and strong staining with 3.

Pearson correlations between SELENOS and VCP/p97 expression with malignant grading and Ki67 values, and between SELENOS and VCP/p97 scores were evaluated by GraphPad Prism 5.0 (San Diego, CA, USA). We indicated *p*-values < 0.05 with *, *p*-values < 0.01 with ** and *p*-values < 0.0001 with ***.

### 2.6. Functional Analysis

Functional analysis on more significant selenoproteins was performed using the STRING tool (https://string-db.org/, accessed on 1 May 2020). The STRING database aims to collect, score and integrate all publicly available sources of protein–protein interaction information and to complement them with computational predictions. For enrichment analysis, STRING implements well-known classification systems, such as Gene Ontology and KEGG, but also offers new classification systems based on high-throughput text-mining, as well as on a hierarchical clustering of the association network itself [38].

### 2.7. Human Interactome Construction

The entire human interactome was constructed using the Cytoscape tool (http://www.cytoscape.org/, accessed on 20 January 2020) [39] and the three following databases that collect physical molecular interactions by experimental studies, as references: (i) IntAct database [40]; (ii) The International Molecular Exchange Consortium (IMEX) [41]; and (iii) the Agile Protein Interactomes DataServer (APID) [42]. The interactions between human proteins, derived from three databases, were downloaded and merged from the Cytoscape tool [39] using the merge function implemented in the software in order to construct the entire human interactome. The obtained network is composed of 38,445 nodes/proteins and 440,390 edges/interactions.

### 2.8. TNBC Network Construction and Selenoprotein Network Analysis

The list of genes involved in TNBC was extrapolated by the GDS2250 dataset (Basal-like breast cancer tumors) in Gene Expression Omnibus (GEO) and TNBC breast cancer TCGA. In detail, the list of over- and down-expressed genes in TNBC samples was selected by calculating the values of fold change obtained on the ratio between the mean expression of each gene in TNBC samples compared to normal tissues. The Cytoscape platform (http://www.cytoscape.org/, accessed on 12 April 2020) was used to map the proteins codified by the selected genes on the entire human molecular interactome and to create the relative interaction network called the “TNBC network”.

We extracted the interaction network between the selenoproteins and the other nodes present in the TNBC network. Some statistical analyses were used to identify the nodes with a large degree and higher number of connections with other nodes in the networks, defined as HUB nodes. In detail, by using the CytoHubba plugin, we evaluated the following topological measures: Node Degree, Edge Percolated Component (EPC), Maximum Neighborhood Component (MNC), Density of Maximum Neighborhood Component (DMNC), Maximal Clique Centrality (MCC) and centrality measures based on shortest paths, such as Bottleneck (BN), Eccentricity, Closeness centrality, radiality, betweenness centrality and stress centrality [43,44].

## 3. Results

### 3.1. In Silico Analysis of Selenoprotein Expression in TNBC Cell Lines

We first analyzed the gene expression profile of all twenty-five selenoproteins in 57 human breast cancer cell lines, 25 TNBC and 32 “non-TNBC”, extracted from the CCLE RNAseq dataset. As reported in Figure 1A by a color scale from green (low levels) to red (high levels), selenoproteins appeared differently expressed in the TNBC subgroup whereas compared “non-TNBC” subgroup. In detail, the analysis of the selenoproteins mean expression profiles evidenced that the gene expression levels of sixteen selenoproteins werehigher and nine selenoproteins were lower in the TNBC subgroup compared to “non-TNBC” subgroup (Figure 1B).

A similar analysis was also performed using the LINCS RNAseq dataset, comprising data related to seventeen selenoproteins in 30 breast cancer cell lines, 20 TNBC and 10 “non-TNBC”. The mean expression profiles showed that the gene expression levels of ten selenoproteins were higher and seven selenoproteins were lower in the TNBC subgroup compared to the “non-TNBC” subgroup (Appendix A). Overall, considering both the CCLE and LINCS datasets, eight selenoproteins were overexpressed (DIO2, GPX1, GPX4, SELENOI, SELENON, SELENOS, TXNRD1 and TXNRD3) and five were down-expressed (DIO1, GPX2, GPX3, SELENOP and SELENOW) in TNBC compared to all other “non-TNBC” breast cancer subtypes.

### 3.2. In Vitro Analysis of Selenoprotein Gene Expression Profile in Breast Cancer Cells

To validate the RNAseq data described above in vitro, we next evaluated the expression of all twenty-five selenoproteins by RT-qPCR in two TNBC cell lines (MDAMB231 and MDAMB468), three estrogen receptor (ER) and progesterone receptor (PR) positive cell lines (HCC1500, MCF7 and MDAMB175), two HER2 positive cell lines (SKBR3 and SUM185), and two ER/HER2 and PR/ER/HER2 positive cell lines (HCC1419 and MDAMB361), first reported as relative expression compared to normal epithelial breast cell line MCF-10 (Appendix A). By this analysis, we found a great variability in the selenoprotein expression among different breast cancer subgroups, but also between cell lines in the same subgroup, accordingly with data from the literature and, probably, to different genetic backgrounds relative to the small number of cell lines analyzed. Thus, to mimic the analysis performed on the CCLE and LINCS datasets, we evaluated the mean expression profiles for each breast cancer subtype and compared to the TNBC cells with all the other breast cancer subtypes indicated as “non-TNBC” cells by clustering analysis (Figure 2A).

In this way, we evidenced that, in agreement with data reported in Figure 1, in the two TNBC cell lines, the expression levels of DIO2, GPX1, GPX4, SELENOS, TXNRD1 and TXNRD3 were higher, whereas those of DIO1, GPX2, GPX3 and SELENOW were lower, compared to “non-TNBC” cells. Notably, among these selenoproteins, considering only the two TNBC cell lines (MDAMB231 and MDAMB468) compared to normal epithelial breast MCF10A cells, we confirmed the overexpression of GPX1, SELENOS, TXNRD1 and TXNRD3 in both cell lines, and GPX4 in one cell line, as well as the downregulation of SELENOW in both cell lines (Appendix A), overall suggesting again that these selenoproteins should play a tumor-related role in TNBC cells.

In this regard, it is important to underline that the expression levels of the selenoprotein GPX1 increased only in TNBC cells, whereas theyhad a decreased trend in other breast cancer subtypes and, in detail, a statistically significant decrease in ER/PR positive cells (Appendix A), in agreement with a recent paper that evidenced a crucial role of GPX1 in the metastatic process of TNBC cells through cell adhesion modulation [45]. Since GPX1 activity was indicated to be correlated to decreased levels of SELENBP1 [46], a protein that has not selenocysteine residues in its sequence but that binds selenium, we also decided to evaluate the gene expression of SELENBP1 by RT-qPCR. As reported in Appendix A, the levels of SELENBP1 resulted to be down-expressed in the TNBC cells compared to other breast cancer cellular subtypes, confirming the inverse correlation between GPX1 and SELENBP1 levels in TNBC cells.

### 3.3. Selenoprotein Gene Expression Evaluation in TNBC Tissues

Next, to further characterize selenoprotein expression and function in TNBC, we evaluated mRNA expression profiles of all twenty-five selenoproteins on thirty human TNBC tissues in comparison with adjacent normal breast tissue counterparts (Table 1) using RT-qPCR. 

The results showed that the mean expression levels of twelve selenoproteins (DIO1, DIO3, GPX1, GPX4, GPX6, MSRB1, SELENOH, SELENOS, SEPHS2, TXNRD1, TXNRD2 and TXNRD3) were statistically higher in TNBC tissues compared to normal counterparts (Figure 2B,C). Interestingly, all these twelve selenoproteins were also overexpressed in both, or at least one, of our TNBC cell lines compared to normal epithelial breast MCF10 cells. Notably, our group has previously evaluated SEPHS2 levels on the same collected TNBC tissues by IHC, confirming that at the protein level this selenoprotein increased in TNBC tissues, compared to their normal counterparts, and increased with the malignant grade [21]. Moreover, in line with the inverse correlation between GPX1 and SELENBP1 highlighted above, we also evaluated the levels of SELENBP1 by RT-qPCR in TNBC tissues, confirming that SELENBP1 levels were statistically lower in TNBC tissues compared to normal counterparts (Appendix A).

Overall, five selenoproteins, GPX1, GPX4, SELENOS, TXNRD1 and TXNRD3, overexpressed in TNBC tissues, were also specifically dysregulated in TNBC cell lines within the CCLE and LINCS RNAseq datasets (Figure 1) and our RT-PCR analysis (Figure 2A).

Finally, we performed functional analyses that suggested common features and functional interactions between those selenoproteins. Indeed, all five selenoproteins are involved in the “antioxidant activity” molecular function, and in two “cellular oxidant detoxification” and “response to oxidative stress” biological processes (Figure 3A and Appendix A). Moreover, the two glutathione peroxidases (GPX1 and GPX4) are involved in three KEGG pathways, “glutathione metabolism”, “thyroid hormone synthesis” and “arachidonic acid metabolism”, and together withTXNRD1, in four REACTOME pathways, “detoxification of reactive oxygen species”, “synthesis of eicosatetraenoic acid”, and metabolism of “nucleotides” and “lipids” (Figure 3A and Appendix A).

### 3.4. Identification of a Strict Correlation between SELENOS and VCP/p97 by Network Analysis

Next, in order to better define the functional involvement of all the selenoproteins in TNBC and to study their correlation with genes modulated in TNBC, we created an interaction network specific for the TNBC subtype. First, we selected the list of over- and down-expressed genes in TNBC samples, compared to the normal subgroup, from GDS2250 and TCGA datasets and analyzed the values of the fold changes, calculated on the ratio between the mean expression of each gene in TNBC samples compared to normal tissues. In this way, we selected 2397 overexpressed and 973 downregulated genes in TNBC. Then, we mapped them on the entire human interactome, created on the basis of the procedure reported in the Methods section, the dysregulated 3370 genes selected above (both the over- and down-expressed genes).

The obtained “TNBC network” comprised 16,446 nodes and 281,792 edges, considering 2nd order interactions, and 4146 nodes and 45,927 edges if we focus our attention on the 1st order sub-network. Then, we extracted the interaction network between selenoproteins and other nodes within the TNBC network, thus obtaining a sub-network composed of 200 nodes and 909 edges, including all the selenoproteins with the exception of DIO1, SELENOW and TXNRD3 (Figure 3B).

A detailed statistical analysis (Appendix A) confirmed that the network was robust and reliable. Indeed, the analysis, based on its centrality and topological properties, evidenced that the network followed the small-world rule with a very shortpath length (3.166) and a density equal to 0.05, thus evidencing good efficacy of the potential connections reported. Moreover, an average number of neighbors equal to 21.55 and the high value of the heterogeneity equal to 1.73, suggested the capacity of the network to contain more correlated nodes, named HUB.

In this way, we selected eight HUB nodes: BRAC1, CDC5L, CUL3, EWSR1, HNRNPA1, TRAF6, YWHAZ and VCP/p97. Focusing on the direct interactions between selenoproteins and the HUB nodes, it is worth noting that only two HUB nodes, CUL3 and VCP/p97, were directly correlated to selenoproteins. In detail, VCP/p97 linked directly SELENOS, DIO2 and CUL3,which, in turn, linked directly to SELENOF. Moreover, VCP/p97 also linked GPX4 through JUN, GPX1 through CEP19, and TXNRD1 through several interactors, such as CAV1, ESR1 HSPA5, ISG15, NTRK1, TP53, TUBA1C or TUBB (see Figure 3C).

Notably, this unbiased interaction network analysis evidenced the strict correlation through VCP/p97 of GPX1, GPX4, SELENOS and TXNRD1, whichare four out of five of the selenoproteins we have reported as dysregulated in TNBC by different approaches.

### 3.5. VCP/p97 and SELENOS Correlated Overexpression in TNBC

Hence, considering that VCP/p97 plays an important and central role as HUB in the interaction sub-network of selenoproteins in the TNBC network, and since it correlates directly with SELENOS [25,26], we decided to evaluate VCP/p97 mRNA and protein expression in TNBC cells. As shown in Figure 4A we evidenced a statistically significant increase of VCP/p97 mRNA expression in the two TNBC cell lines (MDAMB231 and MDAMB468) compared to the normal epithelial breast cell line (MCF-10A).

This overexpression of VCP/p97 was also confirmed at the protein level by western blot analysis and parallel SELENOS overregulation (Figure 4B, Appendix A), the latter confirming the mRNA data presented above.

Next, we evaluated SELENOS and VCP/p97 expression by IHC in the thirty TNBC tissues compared with the normal tissue counterparts (Table 1). SELENOS, which has never, to our knowledge, been evaluated by IHC in TNBC before, was detected in the cytoplasm of both epithelial and myoepithelial cells of the normal component of the mammary gland. All the observed cases are immunoreactive, with a percentage of cells always higher than 30% of the tumoral population, with a positivity predominantly cytoplasmic and sometimes at the cell membrane. All thirty normal counterparts showed a weak positivity of SELENOS. Moreover, we evidenced an increasing intensity in TNBC tissues from grade 2 to grade 3 (Figure 5A). In detail, nine out of ten TNBC tissues with grade 2 showed a moderate positivity and only one had strong positivity; conversely, only two samples of twenty TNBC tissues with grade 3 showed a moderate positivity, while eighteen showed a strong positivity (Figure 5B).

In the case of VCP/p97, the normal component of the mammary gland has a weak and zonal cytoplasmic positivity and this weak staining of normal glandular parenchyma was also observed in the tissue sections that contained both the neoplastic component and rare normal glandular acini in the peritumoral area. Anyhow, all thirty normal counterparts showed a weak positivity. Conversely, in tumor cells, the positivity is predominantly diffuse and cytoplasmic, again with an increasing intensity from the TNBC patients with grade 2 to those with grade 3, as in the case of SELENOS (Figure 5A). In detail, seven out of ten grade 2 TNBC tissues showed a moderate positivity, and only one showed a strong positivity, whereas two samples had a weak staining. On the contrary, fourteen out of twenty grade 3 TNBC tissues showed a strong positivity and five a moderate positivity, whereas only one had a weak staining (Figure 5B).

Moreover, the levels of SELENOS and VCP/p97 in TNBC tissues correlated in a statistically significant manner with *p*-value < 0.0001 *** and Pearson correlation coefficient equal to 0.77 (Figure 5C).

Furthermore, a statistically significant correlation was found between SELENOS and VCP/p97 IHC expression with Ki67 values (*p*-value < 0.05 *), with a strong positivity for both proteins correlating with Ki67 > 50% (*p*-value < 0.001 **) (Appendix A). No correlation was found between SELENOS and VCP/p97 expression with tumor size or lymph node status.

Finally, our results were also confirmed by analyzing the TCGA dataset. In detail, the levels of SELENOS and VCP/p97 in TNBC tissues were higher than in normal samples (with *p*-values < 0.0001 ***) (Figure 6A) and correlated between them with a *p*-value < 0.0001 *** (Figure 6B). Moreover, higher combined gene expression of SELENOS/VCP(p97) in TNBC tissues from the TCGA dataset was associated with poor overall survival (Figure 6C).

## 4. Discussion

As emerged from a recent large descriptive analysis of breast cancer clinical and pathological characteristics in a population-based database [47], for TNBC, due to its heterogeneity, aggressive features and an onset often at a young age are urgently needed robust prognostic and predictive biomarkers, as well as novel molecular targets, particularly in the metastatic setting [1,48].

In this work, we evaluated for the first time by a systematic approach the role of all human selenoproteins as potential prognostic biomarkers and/or their utility as therapeutic targets in TNBC.

Through the integration of several analyses, in silico, interrogating two publicly available large cell lines RNA-seq datasets, in vitro, analyzing by RT-PCR mRNA expression in TNBC cell lines compared to normal breast epithelial cells (MCF-10A), on TNBC tissues, evaluating protein expression by IHC, we studied all twenty-five human selenoproteins and selected GPX1, GPX4, SELENOS, TXNRD1, and TXNRD3, as specifically and statistically significant dysregulated in TNBC.

The glutathione peroxidase (GPX) family has been suggested in the literature as a predictor of response to cytotoxic treatments and prognosis in cancer [49,50]. GPX1 is a selenium-containing enzyme that protects cells against oxidative stress by eliminating hydrogen peroxide and organic hydroperoxides, using reducing equivalents from NADPH via the glutathione–glutathione reductase system. A recently published study demonstrated, using the IHC approach, that high expression levels of GPX1 were associated with shorter overall survival and higher mortality rates in breast cancer patients and played a vital role in the metastasis of TNBC cells by regulating cell adhesion [45]. We also showed, in both TNBC cell lines and tumor tissues, the inverse correlation between GPX1 and SELENBP1, a protein lacking selenocysteine residues in its sequence but binding exogenous selenium, confirming the correlation of these two proteins previously reported in breast cancer cells, in which SELENBP1 was suggested to play a critical role in modulating the extracellular microenvironment by regulating the levels of extracellular GSH [46].

GPX4, another member of the GPX family, is considered a modulator of ferroptotic cancer cell death, driven by lipid peroxide through inhibition of the cystine/glutamate transporter, and is reported to be important for the survival of TNBC cells [51]. GPX4, as GPX1, was found to be highly expressed in different cancer tissues, including breast cancer, and also correlated with shorter patient survival [52,53]. Mechanistically, the role of GPX4 is critical for cell survival since when phospholipid hydroperoxides are not efficiently quenched by this selenoprotein, it triggers a catalytic reaction in the presence of transition metals, such as iron, that eventually causes cell death [54]. Indeed, the development of potent small-molecule inducers of ferroptosis, through GPX4 targeting, has been proposed for cancer therapy [55].

GPX1 and 4 andTXNRD1 and 3 are essential for redox homeostasis and their combined overregulation in TNBC is probably associated with the response of breast cancer cells to increase oxidative stress [56]. TXNRD1 is an activator of thioredoxin, an oxidoreductase targeting cysteine residue of cellular proteins, including redox-sensitive transcription factors, such as NF-κB and p53, playing a critical role in cancer cells and that needs a reducing environment for their DNA binding efficacy [57]. TXNRD1 and thioredoxin interacting protein (TXNIP) were associated with poor breast cancer prognosis, and ERBB2 was suggested to play a role in altering their redox control pathway [57]. Moreover, in breast cancer cells, TXNRD1 overregulation, by inducing epithelial-to-mesenchymal transition (EMT), enhanced invasiveness, thus being considered a candidate for therapeutic targeting [57]. Pan-TXNRD inhibitors, such as auranofin, and specific TXNRD1 inhibitors have been developed in recent years, demonstrating their capacity to induce oxidative stress, to suppress cancer cell growth, and to kill cancer cells, hence confirming their therapeutic anticancer potential [58,59].

Overregulation of TXNRD1 mRNA expression was reported in TNBC patients compared to non-TNBC patients [60]. We have previously reported that TXNRD1 had statistically significant higher levels in breast, head and neck, lung, and prostate cancer patients with poor prognosis [16]. Recently, TXNRD1 redox activity was reported to be higher in TNBC cells compared to non-TNBC cells and the thioredoxin system was correlated with adverse clinical outcomes in TNBC patients [61].

Higher TXNRD3 levels have been reported as indicative of advanced cancer stages in colorectal cancer [62]. However, our study is the first to demonstrate higher levels of TXNRD3 in breast cancer and, in particular, in the TNBC subtype.

SELENOS (also known as SELS, VIMP, SBBI8 or SEPS1) is a selenoprotein predominantly localized in the ER membrane and it is implicated in the retro-translocation of misfolded proteins across the ER membrane back to cytosolic degradation by the ubiquitin-proteasome system [63]. This function is mediated by the binding and recruiting of the AAA+ ATPase VCP/p97, indeed SELENOS is also called VIMP, an acronym for VCP-interacting membrane protein [63]. SELENOS is overregulated in response to ER stress, inflammatory cytokines, and glucose deprivation [64,65]. In breast cancers, where stress arises from hypoxia and nutrient deprivation induced by cytotoxic and endocrine therapeutic interventions, it was reported that chronic activation of the unfolded protein response (UPR), activated by uncontrolled protein synthesis and aggregation of unfolded/misfolded proteins in the ER lumen, is associated with therapy resistance and disease recurrence [66]. Moreover, it was also demonstrated that overexpression of ER to Golgi trafficking gene signature correlated with increased risk of distant metastasis and reduced relapse-free and overall survival in breast cancer patients [67]. SELENOS gene polymorphisms have been correlated to colorectal and gastric cancer development [68,69]. However, no data about the correlation between SELENOS and breast cancer have been reported until now. Hence, our paper is the first that evidenced a putative role of SELENOS in breast cancer and, in particular, in the TNBC subtype.

Most importantly, to our knowledge, this is the first study about SELENOS and VCP/p97 coordinated expression and correlation in TNBC. In detail, our network and bioinformatics studies evidenced and confirmed the strict correlation between SELENOS and VCP/p97 and suggested the important role of VCP/p97 as a HUB node in the interaction with other significant selenoproteins (GPX1, GPX4 and TXNRD1) in the TNBC network. Notably, the correlated expression of SELENOS and VCP/p97 was validated on TNBC tumor tissues, and associated with an increased malignant grade and ki67 values and in TCGA datasets resulted as predictor of poor prognosis.

VCP/p97, in conjunction with a collection of cofactors and adaptors, plays an important role in cellular homeostasis by regulating autophagy, mitochondrial-associated degradation, morphology alteration of nuclear and Golgi membranes, endoplasmic reticulum-associated degradation (ERAD), endosomal trafficking, and chromatin-associated degradation [25,70,71]. VCP/p97 functions in several hallmarks of cancer have been largely reported and specific inhibitors have been developed and tested for cancer treatment [72,73,74,75]. However, only one study reported the evaluation of anti-tumor efficacy of a VCP/p97 inhibitor, DbeQ, in human ER-positive breast cancer cells (MCF-7), showing regulation of cell cycle kinetics through modulation of p21 and p27 protein degradation, and also sensitization of breast cancer cells to several anticancer therapeutics bothin vitroandin vivo [76].

Similarly, in recent years, elevated VCP/p97 expression has been found in various cancers and has been correlated with the progression, prognosis and metastatic potential of esophageal carcinoma, colorectal carcinoma and prostate cancer [77,78]. However, only two studies reported VCP/p97 involvement in breast cancer, evidencing its elevated expression in cancer tissues without focusing on a specific breast cancer subtype [79,80].

Interestingly, both SELENOS and VCP/p97 were found to be secreted proteins. SELENOS was detected in the culture medium of HepG2 liver cells and human serum samples of type 2 diabetes mellitus and atherosclerosis patients [81,82]. Serum VCP/p97 levels were measured in ovarian carcinoma, non-Hodgkin’s lymphoma and breast, colon, pancreatic, lung and prostate cancer patients [83]. Therefore, further studies should focus on the evaluation of these proteins in biological fluids, such as plasma, an ideal source of biomarkers, since it might represent the snapshot of a patient’s pathophysiological state at a given time and might allow dynamic monitoring with insight into the process of spatial and temporal clonal evolution of the tumor, including secondary resistance to treatment, which is denied by the invasiveness of tissue biopsies.

Finally, despite the development of several VCP/p97 inhibitors, mostly targeting catalytic/substrate binding sites [73,74,75,84,85], unfortunately, their development in clinical studies has been disappointing due to adverse effects and lack of efficacy [73,86,87]. In this regard, the development of novel allosteric inhibitors, recently suggested as promising alternatives when resistance to VCP/p97 inhibitors occurs [88], could be directed, on the basis of our findings, to the targeting of SELENOS and VCP/p97 binding site, as an interesting approach for TNBC treatment.

## 5. Conclusions

In the present study, by integrating bothin silicoand wet lab approaches, we demonstrated that the expression profiles of five selenoproteins are specifically dysregulated in TNBC. Most importantly, by bioinformatics analysis, we selected SELENOS and its interacting protein VCP/p97 as inter-related with the others and whose coordinated overexpression is associated with poor prognosis in TNBC.Overall, we confirmed, as previously suggested [12], that only the combined evaluation of some selenoproteins and of HUB nodes could have prognostic value and may improve patient outcome prediction.

We acknowledge the limitations of our study. Although the prognostic performance of the combined elevated expression of SELENOS and VCP/97 was also confirmed in the TCGA dataset, further analyses are warranted and clinical covariates should be evaluated in order to exclude potential confounders. Similarly, we are aware of the limited number of TNBC cell lines and of tumor tissues evaluated in confirmation studies. Nevertheless, we highlighted two mechanistically related novel proteins whose correlated expression could be exploited for a better definition of prognosis, as well as suggested as novel therapeutic targets in TNBC.

## Figures and Tables

**Figure 1 cancers-14-00646-f001:**
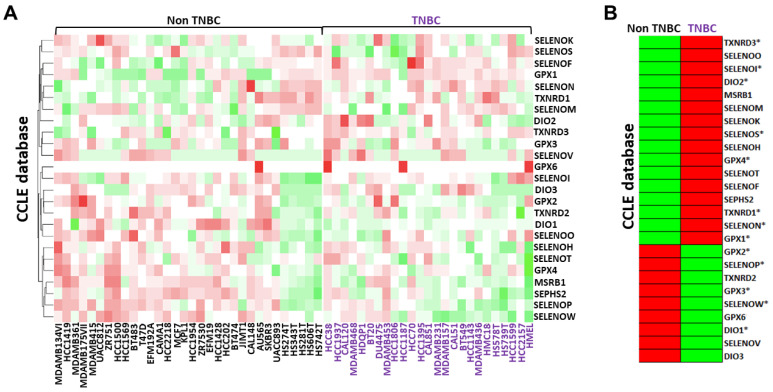
Comparison between the gene expression profiles of each selenoprotein in breast cancer cell lines subdivided between TNBC and “non-TNBC” subgroups and their related mean expression profiles by clustering analysis obtained using RNAseq data in the CCLE database (**A**,**B**). The color scale from green to red refers to lower and higher gene expression levels of selenoproteins, respectively. We evidenced with * the selenoproteins that presented similar expression trends in both the CCLE and LINCS datasets.

**Figure 2 cancers-14-00646-f002:**
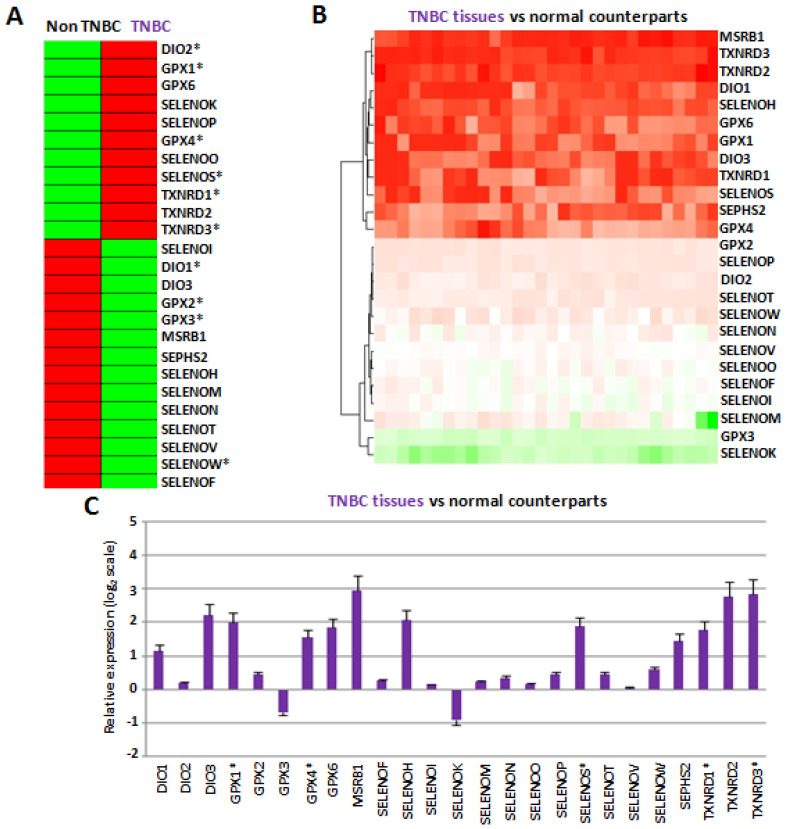
(**A**) Comparison between the mean gene expression profiles of each selenoprotein in TNBC cells and other cellular subtypes obtained considering together ER/PR positive, HER2 positive, ER/HER2 and PR/ER/HER2 positive subtypes, named “non-TNBC” by clustering analysis. Asterisks indicated the selenoproteins that in our cellular studies presented expression trends equal to CCLE and LINCS datasets. (**B**) Comparison between the gene expression profiles of each selenoprotein in thirty TNBC tissues compared to the related normal counterparts. (**C**) Mean fold changes of gene expression levels for each selenoprotein in all TNBC tissues compared to the all normal counterparts, evaluated by the 2^−ΔΔCq^ method and reported as log_2_ scale. We consider values higher and lower than +1 and −1, respectively, statistically significant. We evidenced with ^‡^ the selenoproteins that presented similar expression trends in both TNBC cells and tissues, and CCLE and LINCS datasets.

**Figure 3 cancers-14-00646-f003:**
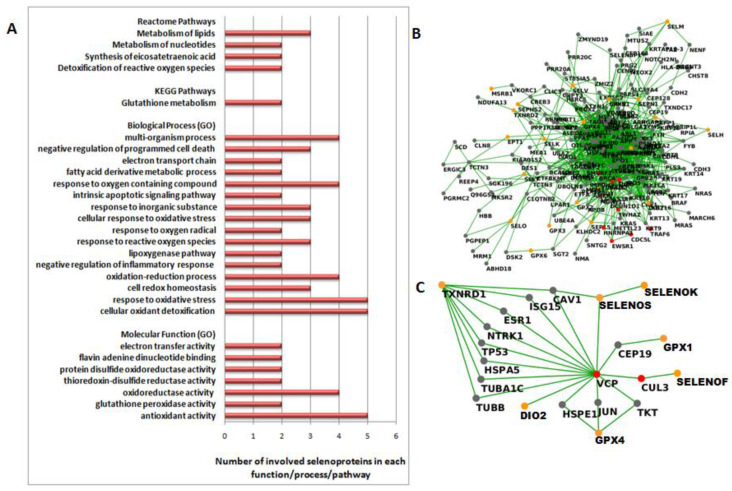
(**A**) Functional analysis performed by the STRING tool (https://string-db.org/, accessed on 1 May 2020) on the five more significant selenoproteins (GPX1, GPX4, SELENOS, TXNRD1 and TXNRD3). The bars indicate the number of selenoproteins involved in each Molecular Function and Biological Process Gene Ontology (GO), and Reactome and KEGG Pathways. (**B**) First-order interaction network obtained by extracting the selenoproteins from the TNBC network. We report selenoproteins in orange, HUB nodes in red and other nodes in gray. (**C**) Sub-network related to the links between more significant selenoproteins and two HUB nodes, CUL3 and VCP/p97. We report DIO2, GPX1, GPX4, SELENOF, SELENOK, SELENOS and TXNRD1 in orange, CUL3 and VCP/p97 in red and other nodes in gray.

**Figure 4 cancers-14-00646-f004:**
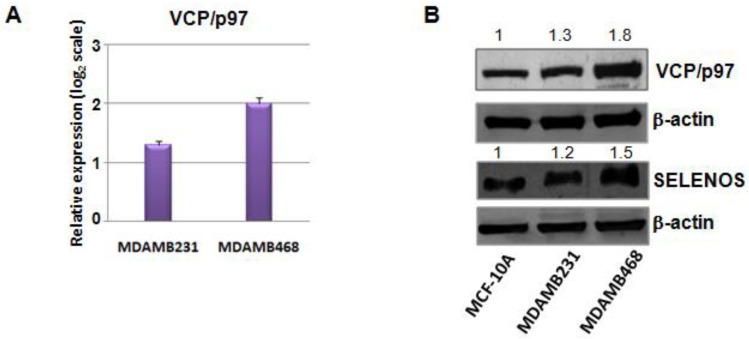
(**A**) Fold changes of expression levels for VCP/p97 mRNA in two TNBC cell lines (MDAMB468 and MDAMB231) compared to the non-cancerous MCF-10A cells, evaluated by the 2^−ΔΔCq^ method and reported as log_2_ scale. We consider values higher than +1statistically significant. (**B**) Western blot validation of VCP/p97 and SELENOS expression on MCF-10A, MDAMB231 and MDAMB468 cell lines. β-actin was used as a loading control. The numbers indicate the quantifications performed for VCP/p97 and SELENOS by IMAGEJ software.

**Figure 5 cancers-14-00646-f005:**
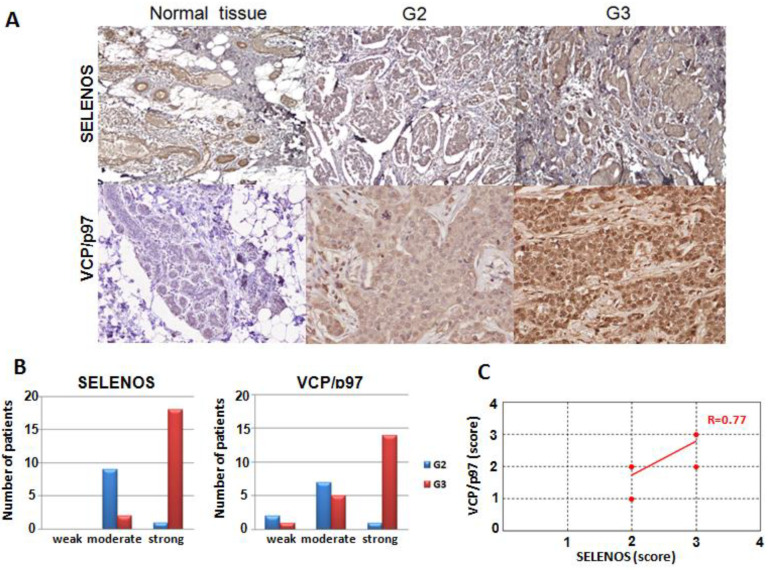
(**A**) Immunohistochemical expression of human SELENOS and VCP/p97 at 200× magnification in normal mammary tissues and TNBC patients with grade 2 and 3. In grade 2 (G2) and grade 3 (G3) tissues the neoplastic cells form solid clusters separated by bands of connective tissue. (**B**) Correlation between SELENOS and VCP/p97 expression and tumor grading. (**C**) Pearson correlation between SELENOS and VCP/p97 scores obtained by IHC (R, correlation coefficient).

**Figure 6 cancers-14-00646-f006:**
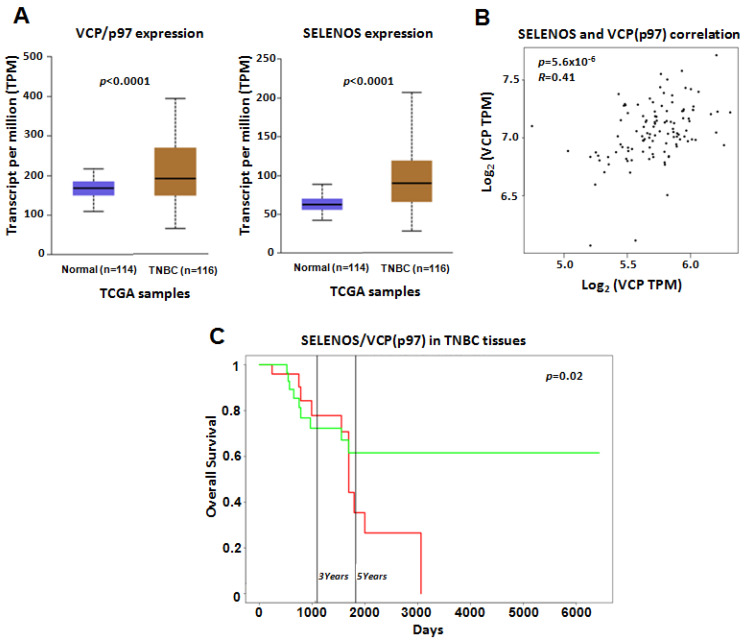
(**A**) VCP/p97 and SELENOS gene expression reported as transcript per million (TPM) in the TCGA dataset. We report the *p*-values indicating the statistically significant difference between normal and TNBC samples. (**B**) Pearson correlation between SELENOS and VCP/p97 expression in TNBC samples from the TCGA dataset. Correlation coefficient (R) and *p*-value are reported. (**C**) Kaplan–Mayer curves showing the correlations between overall survival (expressed in percentage) and the combined gene expression of SELENOS/VCP(p97) in TNBC tissues in the TCGA dataset, by PROGgeneV2 online tool. High and low expression of SELENOS/VCP(p97) are reported by the red and green curves, respectively.

**Table 1 cancers-14-00646-t001:** Clinic-pathological assessment of patients.

Patient Characteristics	Number of Patients
Age (mean ± st.dev.)	52.6 ± 13.5
primitive TNBC	30
histotype	29: no special type; 1: metaplastic
grading	grade 2: 10; grade 3: 20
Ki67 (range, mean ± st.dev)	grade 2: 10–50%, 28.9 ± 13.3%
	grade 3: 25–70%, 44.9 ± 15.8%
Lymphnode status (pN)	grade 2: 8pN0, 2pN1a
	grade 3: 12pN0, 1pN1, 4pN1a, 1pN1c, 1pN2, 1pN2a
Tumor size (pT)	grade 2: 1pT1a, 2pT1b, 3pT1c, 3pT2, 1pT3
	grade 3: 9pT1c, 9pT2, 1pT3, 1pT4a
Status	23 (Live); 7 (Dead)

## Data Availability

The data presented in this study are available on request from the corresponding author. The data are not publicly available due to privacy.

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
