# Peer review of "An Integrated In Silico, In Vitro and Tumor Tissues Study Identified Selenoprotein S (SELENOS) and Valosin-Containing Protein (VCP/p97) as Novel Potential Associated Prognostic Biomarkers in Triple Negative Breast Cancer"

_cancers, 2022, doi:10.3390/cancers14030646_

Round 1

Reviewer 1 Report

1- Structure the Abstract in Sections (Background, Methods, Results, Conclusion). Please make sure that the general Methods of the study are clear to the readers in the Abstract. 

2- In the Manuscript, start the Methods Section with a general description of the study, so the general methods of the study are clear to the reader

3- Cite and discuss this article:

Sisti A, Huayllani MT, Boczar D, Restrepo DJ, Spaulding AC, Emmanuel G,
Bagaria SP, McLaughlin SA, Parker AS, Forte AJ. Breast cancer in women: a
descriptive analysis of the national cancer database. Acta Biomed. 2020 May
11;91(2):332-341. doi: 10.23750/abm.v91i2.8399. PMID: 32420970; PMCID:
PMC7569667.

Author Response

Reply to Reviewer 1

  • Structure the Abstract in Sections (Background, Methods, Results, Conclusion). Please make sure that the general Methods of the study are clear to the readers in the Abstract.

As suggested by this reviewer, we modified the Abstract subdividing it in Sections (Background, Methods, Results, Conclusion) clarifying the Methods used in this paper

  • In the Manuscript, start the Methods Section with a general description of the study, so the general methods of the study are clear to the reader

As suggested by this reviewer, we inserted at the start of Methods a general description of the study to make clear to the readers the general methods of the study

3- Cite and discuss this article:

Sisti A, Huayllani MT, Boczar D, Restrepo DJ, Spaulding AC, Emmanuel G, Bagaria SP, McLaughlin SA, Parker AS, Forte AJ. Breast cancer in women: a descriptive analysis of the national cancer database. Acta Biomed. 2020 May 11;91(2):332-341. doi: 10.23750/abm.v91i2.8399. PMID: 32420970; PMCID: PMC7569667.

As suggested by this reviewer, we cite and discussed this article at the beginning of the Discussion Section.

Reviewer 2 Report

Overall, interesting study and findings.

In the introduction you indicate "The role of selenoproteins in carcinogenesis... remain ambiguous..." yet your findings seem to imply that most if not all of the selenoproteins promote cancer initiation and/or progression. Please explain why this might be true.

Author Response

Reply to Reviewer 2

Overall, interesting study and findings.

In the introduction you indicate "The role of selenoproteins in carcinogenesis... remain ambiguous..." yet your findings seem to imply that most if not all of the selenoproteins promote cancer initiation and/or progression. Please explain why this might be true.

We thank this reviewer for his/her comment, and, consequently we better explained in the introduction why, although the selenoproteins seems to be involved in cancer initiation and/or progression, it is not clear how and if they can be used as clinical diagnostic/prognostic markers.

Reviewer 3 Report

The paper named “An Integrated in silico, in vitro and Tumor Tissues Study Identified Selenoprotein S (SELENOS) and Valosin-containing Protein (VCP/p97) as Novel Potential Associated Prognostic Biomarkers in Triple Negative Breast Cancer “ is a interesting work were author make a well defined integrating cellular, tissue and bioinformatics studies to determinate  if selenoprotein S and valosin-containing protein can be considered as a new potential associated prognostic biomarkers and therapeutic targets in TNBC.

Only some minor questions are required

  • Authors have made qPCR determinations in paraffin tissues. The paraffin extraction is sometimes complex and can have interferences in the qPCR even using commercial kits specifics for paraffin RNA extraction. Have authors make some analysis in fresh/ freezing tissues?
  • In material and methods section antibodies dilutions and conditions for western blot analysis have been missing.
  • In page 7 line 305 author say that they determinate seventeen selenoproteins hoverer in figure 1 there are eighteen.
  • In point 3.2 authors only used 2 TNBC cell lines when they evaluated the mean expression profiles for each breast cancer subtype and compared the TNBC cells with all the other breast cancer subtypes indicated as“non-TNBC” cells by clustering analysis 333 (Figure 2A). It is possible that the difference in number of samples between TNBC and non-TNBC can modified the correct data? It is possible to use more TNBC cell lines in this analysis?
  • Line 339 repeats the same as indicated at the end of the figure caption.
  • In paragraph between lines 356-359 authors say “On this regard, it is important to underline that the expression levels of the seleno-protein GPX1 increased only in TNBC cells, whereas decreased in other breast cancer subtypes (Figure S2), in agreement with a recent paper that evidenced a crucial role of GPX1 in the metastatic process of TNBC cells through cell adhesion modulation [45].” However in figure S2 the decrease of GPX1 expression is only seen in ER/PR positive cell lines.
  • In figure 4B the numbers that indicate the quantification may be shown as an expression graph.
  • What do the authors think about the low value they obtain in the correlation coefficient in Figure 6B?. The values between the samples are very heterogeneous.
  • Have author do same experiment in tissues from TNBC patients submitted to surgery and Neoadjuvant chemotherapy? The levels of selenoproteins can give a therapy response data?
  • In figure S1 GPX6 is missing in figure S1A
  • In figure S2 how authors calculate the log 2 value? This mean first they perform the qPCR to obtain the AACq for each cell line and them they make a media value from all the cell lines prior to calculate the log 2 value?

Author Response

Only some minor questions are required

Authors have made qPCR determinations in paraffin tissues. The paraffin extraction is sometimes complex and can have interferences in the qPCR even using commercial kits specifics for paraffin RNA extraction. Have authors make some analysis in fresh/ freezing tissues?

We agree with the comment of this reviewer, however, the quality of the RNA extracted from FFPE was good and allowed our analysis. Unfortunately, we had not fresh/ freezing tissues for the examined patients.

In material and methods section antibodies dilutions and conditions for western blot analysis have been missing.

We inserted in the western blot analysis paragraph in material and methods section that the antibodies were diluited 1:1000 in 5% nonfat dry milk, 1X TBS and 0.1%Tween at 4°C.

In page 7 line 305 author say that they determinate seventeen selenoproteins however  in figure 1 there are eighteen.

We thank this reviewer for his/her comment. There was a mistake in Figure S1B. The correct number is seventeen; thus we modified this point in the text.

In point 3.2 authors only used 2 TNBC cell lines when they evaluated the mean expression profiles for each breast cancer subtype and compared the TNBC cells with all the other breast cancer subtypes indicated as“non-TNBC” cells by clustering analysis 333 (Figure 2A). It is possible that the difference in number of samples between TNBC and non-TNBC can modified the correct data? It is possible to use more TNBC cell lines in this analysis?

Of course, as pointed out by the reviewer the limited number of cell lines in the “wet” experiments could represent a bias, indeed we clearly confirm this limination in the conclusion of our original manuscript. Unfortunately, we had not other TNBC cell lines available in our laboratory and could not expand our analysis. However, we are confident on the overall results obtained due to the the concordance of the “wet” results with those derived from in silico analysis on a large number of cell lines, together with other analyses on TNBC tissues and TCGA datasets.

Line 339 repeats the same as indicated at the end of the figure caption.

We modified this point in the text.

In paragraph between lines 356-359 authors say “On this regard, it is important to underline that the expression levels of the seleno-protein GPX1 increased only in TNBC cells, whereas decreased in other breast cancer subtypes (Figure S2), in agreement with a recent paper that evidenced a crucial role of GPX1 in the metastatic process of TNBC cells through cell adhesion modulation [45].” However in figure S2 the decrease of GPX1 expression is only seen in ER/PR positive cell lines.

We modified this phrase evidencing that “GPX1 had a decreased trend in other breast cancer subtypes and, in detail, a statistically significant decrease in ER/PR positive cells”

In figure 4B the numbers that indicate the quantification may be shown as an expression graph.

We inserted new Figure S5 with the quantification shown as an expression graph.

What do the authors think about the low value they obtain in the correlation coefficient in Figure 6B?. The values between the samples are very heterogeneous.

Figure 6B has been obtained from the analysis of correlation between SELENOS and VCP expression by TCGA dataset containing more than 100 TNBC patients. TNBC is a very heterogenous breast cancer subtype. Therefore, the low correlation coefficient is certainly due to the heterogeinty between the samples. However, the p-value <0.05 is index of a statistically significant correlation between SELENOS and VCP expression.

Have author do same experiment in tissues from TNBC patients submitted to surgery and Neoadjuvant chemotherapy? The levels of selenoproteins can give a therapy response data?

We did not experiments in tissues from TNBC patients submitted to surgery and Neoadjuvant chemotherapy. We thank this reviewer for his/her interesting suggestion, and furher studies will regard this type of esperiments.

In figure S1 GPX6 is missing in figure S1A

As indicated above, there was a mistake in Figure S1B. We inserted the correct figure.

In figure S2 how authors calculate the log 2 value? This mean first they perform the qPCR to obtain the AACq for each cell line and them they make a media value from all the cell lines prior to calculate the log 2 value?

We calculalted log2 2-DDCt for each selenoprotein in each cell line

Reviewer 4 Report

Manuscript "An integrated in silico, in vitro and tumor tissues study identified selenoprotein S (SELENOS) and valosin-containing protein (VCP / p97) as novel potential associated prognostic biomarkers in triple negative breast cancer" is an interesting research paper. It also has a potential role in the treatment of patients with triple-negative breast cancer as a prognostic marker.
All parts of the manuscript have been compiled appropriately. The methods used in the research and the obtained results were described in detail.
Due to the interesting topic, the proper selection of molecular, genetic, immunohistochemical and statistical methods and the potential possibility of using research in TNBC therapy, I propose to publish the manuscript "An integrated in silico, in vitro and tumor tissues study identified selenoprotein S (SELENOS) and valosin- containing protein (VCP / p97) as novel potential associated prognostic biomarkers in triple negative breast cancer "in CANCERS in the present form.

Author Response

We thank this reviewer for his/her possitive comments

Round 2

Reviewer 1 Report

1- At the beginning of the Discussion Section, you refer to the study by Sisti et al, but you cite it as reference number 48, instead in the reference list it is number 47. Please fix that.